# A protocol for the evaluation of a wearable device for monitoring of symptoms, and cueing for the management of drooling, in people with Parkinson's disease

**Lorelle Dismore**[1], **Kyle Montague**[2]*, **Luis Carvalho**[2], **Tiago Guerreiro**[3], **Dan Jackson**[4], **Yu Guan**[4], **Richard Walker**[1]

**1** Northumbria Healthcare NHS Foundation Trust, Innovation, Research and Development, North Tyneside General Hospital, North Shields, United Kingdom, **2** Computer & Information Sciences, Northumbria University, Newcastle upon Tyne, United Kingdom, **3** LASIGE, Faculdade de Ciências, Universidade de Lisboa, Portugal, Portugal, **4** Open Lab, School of Computing, Newcastle University, Newcastle upon Tyne, United Kingdom

* kyle.montague@northumbria.ac.uk

**Data Availability Statement:** No datasets were generated or analysed during the current study. All

## Abstract

Drooling is a common symptom of Parkinson's Disease (PD) experienced in up to 70% of people with PD (PwP). Drooling can be a major problem in PwP leading to adverse physical and psychosocial issues. Current medical treatments decrease the production of saliva, whereas the problem is due to decreased swallowing frequency, not over production of saliva. Such treatments are problematic as saliva is essential for good oral health. Therefore, non-invasive treatments options such as behavioural cueing methods are recommended. A wrist-worn device delivering haptic cueing has been demonstrated to be an effective treatment method to increase swallowing frequency and a socially acceptable solution for PwP. However, the device had limited functionality and was tested on a small sample size over a short period of usage. Further work is required to understand the real-world behaviours and usage of the intervention to understand the longer-term effects with a larger sample size. This research will deploy CueBand, a discrete and comfortable wrist-worn device designed to work with a smartphone application to support the real-world evaluation of haptic cueing for the management of drooling. We will recruit 3,000 PwP to wear the device day and night for the intervention period to gain a greater understanding of the effectiveness and acceptability of the technology within real-world use. Additionally, 300 PwP who self-identify as having an issue with drooling will be recruited into an intervention study to evaluate the effectiveness of the wrist-worn CueBand to deliver haptic cueing (3-weeks) compared with smartphone cueing methods (3-weeks). PwP will use our smartphone application to self-assess their drooling frequency, severity, and duration using visual analogue scales and through the completion of daily diaries. Semi-structured interviews to gain feedback about utility of CueBand will be conducted following participants completion of the intervention.

relevant data from this study will be made available upon study completion.

**Funding:** This study has been funded by Parkinson's UK charity [https://www.parkinsons.org.uk/]. The grant holder is Professor RW, grant reference H-2002. The study will be advertised on the Parkinson's UK website to aid recruitment. Parkinson's UK were not involved in the preparation of the manuscript.

**Competing interests:** The authors have declared that no competing interests exist.

## Introduction

Sialorrhea, also known as drooling or ptyalism, is reported to be a common symptom of Parkinson's Disease (PD), affecting up to 70% of people with PD (PwP). Sialorrhea increases in both frequency and severity as the disease progresses [1]. Saliva plays an important role in oral health maintenance, mastication, deglutition, the initiation of the digestive process, and supports clear speech. Impaired flow or consistency of saliva exposes people to risks of lowered resistance to infection, depressed oral health, impaired bolus formation and transportation and implications for digestion. Consequences include dry mouth, ulceration, tooth decay, gingivitis, candidiasis, halitosis and perioral dermatological issues [2–4].

Drooling can be a major problem in PwP due to decreased automatic swallowing [5–7], and cognitive impairment can influence drooling likelihood and severity [8]. These changes may not be directly associated with cognitive status so much as associated changes to attention, particularly when people are multi-tasking and concentrating on other things such as watching television [1, 9, 10]. PwP do not produce more saliva than individuals not living with PD; however, when automatic swallows don't occur saliva pools in the mouth leading to drooling which can be very embarrassing and restricts social life of PwP. The secondary effects from drooling (e.g., odour, stained clothes, constant wiping) are socially undesirable in many societies and presence of sialorrhea may bring repercussions for psycho-social health of the person who drools and added burden for the carer (e.g., washing clothes; restricted social life) [1].

Current treatments decrease the production of saliva either with oral agents such as glycopyrronium, or via Botox injections into salivary glands which need to be repeated on a 3-monthly basis. These treatments are problematic, as it is well known that saliva is essential for good oral health. Impaired production or loss of saliva through drooling therefore exposes individuals to a range of negative effects including major health and psychosocial issues. NICE guidelines [11] on the treatment of drooling problems recommend that nonpharmacological therapy (such as behavioural stimulus methods) should be preferred over pharmacological or surgical therapy. To improve patient outcomes, recent and emerging evidence supports the use of digital therapeutics to address both motor and non-motor aspects of PD [12]. Digital therapeutics are defined as 'delivering evidence-based therapeutic interventions to patients that are driven by software to prevent, manage, or treat a medical disorder or disease [13]. Digital therapeutics are a scalable mechanism that overcome the barriers and shortcomings associated with traditional in-clinic care by delivering personalised evidence-based treatment remotely at a time and place most convenient and beneficial to the patient [12]. Cueing has been employed to successfully improve aspects of impaired activities in PD, such as gait. Our previous research has piloted this approach using a wrist-worn device to provide haptic cues for automatic swallowing [14].

### The Cue Band project

Fig 1 outlines the schedule of enrolment, interventions and assessments for the Cue Band study. The project involves the evaluation of CueBand, an open-source wrist-worn technology for PwP, which has specifically been designed to provide cueing for swallowing in addition to the typical activity monitoring we see from mainstream fitness trackers. Our companion smartphone applications for both Apple iOS and Google Android devices will be published to their perspective app stores and listed free of charge. Upon downloading and installing the application, individuals will be invited to participate in our remote study of CueBand. We aim to recruit 3,000 participants and individually mail them a CueBand device. The smartphone apps will provide our participants with complete control over CueBand's cueing mechanisms,

| TIMEPOINT** | Enrolment<br>-$t_1$ | Allocation<br>0 | STUDY PERIOD — Interventions and Assessments<br>3 weeks | Wash out 2 weeks | 3 weeks | 3 weeks | Follow ups | etc. | Close-out<br>$t_x$ |
|---|---|---|---|---|---|---|---|---|---|
| **ENROLMENT:** | | | | | | | | | |
| **Eligibility screen** | X | | | | | | | | |
| **Informed consent** | X | | | | | | | | |
| *Included in the overall study* | X | | | | | | | | |
| **Allocation to intervention** | | X | | | | | | | |
| **INTERVENTIONS:** | | | | | | | | | |
| *[Intervention Group A]* | | | ◆—————————————◆ | | | | | | |
| *[Intervention Group B]* | | | ◆—————————————◆ | | | | | | |
| *Optional phase* | | | | | | ◆————◆ | | | |
| **ASSESSMENTS:** | | | | | | | | | |
| *[Baseline Daily Diaries]* | | X | | | | | | | |
| *[Daily Diaries]* | | | X | X | X | X | X | etc. | |
| *[Optional interview]* | | | | | | | X | etc. | X |

**Fig 1. Schedule of enrolment, interventions, and assessments.**

including being able to define a 7-day cueing schedule, personalising the intensity of the haptic feedback and the frequency of the cueing interval. The smartphone applications will allow participants to maintain daily diaries and self-report on their drooling frequency, severity, and duration–all of which can be remotely and securely shared with the researchers. We will invite a subset of 300 PwP experiencing drooling (based on a self-reported questionnaire) to participate in our haptic cueing intervention study that will compare wrist-worn, and smartphone delivered haptic cueing (See Groups A & B in Fig 2). Participants in the formal cueing intervention study will not have the control over which cueing method they use (i.e., cueing from Smartphone, or cueing from Cue Band device), as this will be dictated by the study protocol. However, the remaining 2,700 participants in our *free-living* (see Group C in Fig 2) group will have full control of the application and cueing method, with no imposed usage protocol. We are asking these participants to use the technology as they see fit to best support their cueing needs. This group will also be invited to provide anonymous usage data and feedback on the technology designs.

## Materials and methods

In a proof of concept and feasibility pilot [14], employing a wrist-worn tactile cue (PD-CUE) to increase swallowing regularity, 22 from 28 participants found positive benefits, with significant overall group differences on visual analogue scale self-rating of drooling frequency and severity pre-post the 4-week intervention. Early indications are that devices prove highly

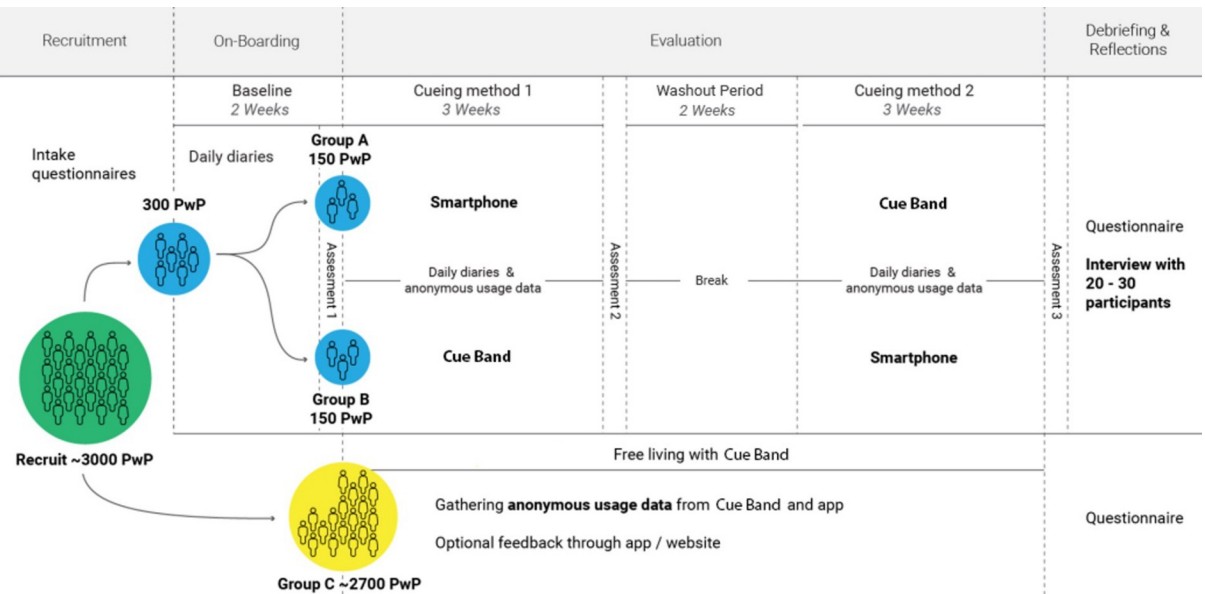

**Fig 2. Overview of the cueing study recruitment and design; including a breakdown of the study activities.** NB for the cueing study we will also recruit from patients under the care of the Northumbria Healthcare NHS Foundation Trust (NHCFT) Parkinson's Service.

successful for many people with PD, but not everyone. Further work is required to test innovative cueing methods on larger populations.

## CueBand

CueBand is a small wrist-worn device capable of providing programmable haptic vibrations to the wearer. In addition to the controllable vibration motor, the device has a 3-axis accelerometer (capable of a signal response up to 200Hz), an onboard heart rate sensor, 1.3inch capacitive touch display, supports Bluetooth 5.0 BLE communication, and is powered by a 180mAh battery. These hardware specifications are comparable with mainstream fitness trackers; however, it is the onboard software (i.e. firmware) that sets CueBand apart from consumer devices. Our custom firmware allows the device to be updated or reprogrammed in seconds 'over-the-air' via a Bluetooth device such as a smartphone or personal computer. This means that it is possible to change and update the functionality of any CueBand device remotely via the individual's smartphone. There are no restrictions on the amount of times that the device can be updated or reprogrammed, making this an exciting device for future healthcare interventions.

## Overall study design

The CueBand devices and the mobile applications for iOS and Android smartphones have been developed–all of which have been developed to be open-source technologies that will be made publically available at the end of the project. Ethical approval has been granted from Northumbria University ethics review board (#32968) for this component of the study. The CueBand evaluation phase will involve the recruitment of 3,000 PwP to wear the device throughout an eight-week evaluation. Participants will be asked to wear the CueBand continuously or as much as possible during this time, giving us a greater understanding of the effectiveness and acceptability of the technology within real-world use. All 3,000 participants in the main study will be posted a free CueBand device, which they can keep and continue to use

beyond the study. Upon receiving their CueBand device, participants will be instructed to activate and pair the device to their smartphone using our companion application. Our apps will automatically scan for and detect the CueBand via Bluetooth communication and pair with the device, at which point the firmware running on CueBand will be automatically updated from our servers and the participant can begin to setup and personalise their cueing schedule and preferences.

## Objectives of the evaluation phase

1. Launch CueBand and begin user evaluation with 3,000 PwP.

2. Publish the open-source CueBand hardware and software for public use.

3. Disseminate project outcomes and publish datasets.

**Procedure of the main study.** The study will be advertised through Parkinson's UK and Special Interest Group on Parkinson's and Technology (SIGPAT) research networks and mailing list, through a project website (https://cue.band/) and through the two participating National Health Service (NHS) organisations. Participants will be able to register their interest in taking part in the research via the project website. During registration participants will be provided with information about the research and they will agree to be contacted about the intervention study. Following registration, the research team will send out instructions to download the CueBand app to their smartphone device to allow them to enrol and consent to participate in the study and request their free CueBand device to be posted to their home address. Participants will be allowed to keep and continue to use the CueBand device and mobile apps beyond the study–we will disable the sharing of research data at the end of our study, making any data capture and diary entries for personal use only by the PwP.

Data from the CueBand device, mobile application and website will be stored securely in a cloud hosting service (e.g Digital Ocean, AWS), only directly accessible by the researchers in this study and according to the protocols for research data storage in Northumbria University. We will comply with GDPR rules - all data in this category will be encrypted and available for the participants to download or be erased and forgotten.

Data regenerated from interviews, workshops, surveys, and questionnaires will be stored securely within Northumbria University according to the protocols for research data storage within this institution. We will comply with GDPR rules - all digital data in this category will be stored in university encrypted OneDrive and following the interview in standalone, password-protected devices.

Confidential data will be disposed of in accordance with the existing systems and protocols within the university. Data will be erased at the end of the project duration.

## Cueing study design

We will take a mixed-methods approach. Our recruitment strategy and study design are illustrated in Fig 2. We will conduct a comparison study with the current state of the art assistive device–smartphone app providing haptic vibration notifications to prompt for swallowing. Using quantitative methods, we will explore the potential effects of the cueing method (i.e., CueBand wearable vs. smartphone only) on the perceived drooling severity and frequency. It is estimated there are 145,000 PwP in the UK, of whom up to 70% experience symptoms of oropharyngeal dysphagia, giving a population of 118,900. We will recruit 300 PwP (95%, CI = 5.65%) to participate in our ten-week cueing intervention study in which we are using a within-subjects design.

**Procedure of the cueing method intervention.**    Three hundred PwP of the 3,000 will be invited to receive the cueing method intervention (CueBand wearable device and smartphone only). This will include group A (150 PwP) and group B (150 PwP) (Fig 2), where the participants will be assigned in turn from a sequence of 300 entries of equal allocations to each condition, which was shuffled into a random order by a computer using a cryptographic random number source. Group C (Fig 2), the 2700 PwP will only be providing anonymous usage data (i.e., cueing schedule, usage activity). They will also have the option to provide feedback and suggestions through the mobile app. Group A and group B will take part in an eight-week cueing method intervention and will experience both interventions. The intervention order will be counter-balanced to avoid learning effects or additional bias and will be based on a randomised allocation. They will be asked to use the intervention for a period of three weeks, while maintaining the daily diaries and self-reporting. After three weeks, the participants will have two weeks of no intervention–the cueing methods will be deactivated for this time period. Then the participants will receive the alternative intervention for three weeks. The CueBand method will be disabled when users are within the Smartphone only intervention period, and vice-versa for the CueBand intervention period. Prior to receiving the intervention, participants will complete a two-week baseline during the on boarding phase, whereby they will complete a daily diary (drooling severity chart) and self-reporting without any intervention via the App. Participants will be required to maintain the daily diary while receiving the intervention (six weeks in total). They will self-report on their swallowing severity, frequency and duration using visual analogue scales, placing a cross on a 100-millimetre (mm) line, (0 mm being no problem) and (100 mm being as bad as can be) (12); as well as provide any additional comments or reflections on their experiences that day. Participants will be sent reminders via the App to compete the study assessments.

A subgroup who express an interest, will enter a 3-week phase at the end of the interventions, during which they receive neither intervention to see if there is a carry-over effect. They will then have the same follow-up assessments as at the end of the intervention period prior to exiting the study.

Following the intervention phase of our study, participants will be asked to complete a questionnaire about their experience and preferences regarding both cueing methods (CueBand and Smartphone only). Participants will complete an adapted System Usability Scale [15]. Participants will be allowed to retain the CueBand wearable device beyond our evaluation; however, we will disable any data sharing with the research team, unless they give permission for any further time limited data to be collected. As part of the debrief phase we will assist the participants in setting up their preferred prompting method (CueBand wearable or Smartphone only), or disable and remove the smartphone applications if they no long wish to use them.

We will aim to recruit until data saturation from the intervention study (Groups A & B) to participate in a subsequent semi-structured interview (in-person, via Telephone or Online audio/video call). Through qualitative methods, we hope to understand daily experiences of PwP with symptoms of drooling and their opinions on the acceptability and feasibility of personal assistive technologies to cue for swallowing.

**Protocol registration.**    This study has been registered on the International Standards Randomised Controlled Trial Number (ISRCTN), registration number 59284050.

**Participants.**    PwP who meet the criteria for the cueing study will be recruited via Parkinson's UK & Digital Parkinson's research networks and mailing lists. PwP will also include patients under the care of the Northumbria Healthcare NHS Foundation Trust (NHCFT) PD Service and the North Cumbria Integrated Care Trust (NCIC) PD Service. If patients are able to complete the online application to participate in the study we will assume that they have

"sufficient cognitive function and manual dexterity". For those who are recruited in clinic we will not include those people who do not appear to have "sufficient cognitive function and manual dexterity", and they would, in any case, be guided to complete the application themselves. This study has received favourable opinion from Newcastle North Tyneside 1 Research Ethics Committee, IRAS ID 305798.

**Eligibility criteria.**

- PwP experiencing symptoms of oropharyngeal dysphagia or self-reporting as having difficulties with drooling and swallowing.

- PwP that own an iOS or Android smartphone device (to support the research engagement). Their smartphone must support Bluetooth BLE 4.0+ to communicate with the band (relatively standard in smartphones since 2010).

- PwP willing and able to provide consent to participate.

**Exclusion criteria.**

- Participants will need to have sufficient cognitive function and manual dexterity with/ without assistance from carers.

We are building on top of the existing PINE64 open-source PineTime Smartwatch project (https://wiki.pine64.org/wiki/PineTime), by developing our own custom version of the InfiniTime firmware (https://github.com/JF002/InfiniTime). We have extended the existing smartwatch functionality (e.g., telling the time, and tracking physical activity) to enhance device security, support time-based cueing schedules and research data logging. Our custom firmware is compatible with all existing PineTime devices, and we will work with the open-source community to provide continued support and combability for future versions.

## Data collection assessments

Participants will initially complete a basic demographic information (e.g., age, gender) and PD related information (e.g., time since diagnosis)–these data will be used to support the analysis and reporting of our study outcomes.

**Daily diaries.**  We will capture *formal assessments* of the individual's symptoms for two weeks prior to the start of the intervention and before and after receiving each cueing method intervention (Fig 2) captured through the mobile app interface. *Formal Assessments* will comprise of questions from the Radboud Oral Motor Inventory for PD subscale for saliva (ROMP-S) [16], the drooling item in the UPDRS 2.2 [17], new non-motor symptom questionnaire (NMSQ) [18], and the Parkinson's Disease Questionnaire (PDQ-8) [19]. Participants will also be asked if they have used the cueing app previously and, if so, what their experience of it was. Participants will be asked to complete daily assessments via dairy entries using Likert-Scales reporting on frequency: to indicate the number of separate episodes of drooling they experienced during that day, duration: to indicate how long those individual episodes lasted and severity of drooling: to indicate the perceived impact of the episodes. "Participants will be encouraged to report any safety issues or adverse events as soon as possible via the app or, if necessary, by phone or email. These will be followed up as soon as possible and, if there are implications for other participants, these will be shared via the app".

**Anonymous usage data.**  All 300 participants will be invited to share their anonymous usage data from the CueBand and companion mobile app. Their data will comprise of their cueing schedule (pre-programmed daily schedule for the cueing periods) and cueing usage (any additional manual activation or deactivation of the cueing beyond the pre-programmed

behaviours). Other data will include physical activity (time-banded levels of physical activity and movement during awake periods) and sleep behaviours (time-banded measurements of physical movements during periods of sleeping).

**Additional comments and reflections.** Finally, participants will have the option to anonymously provide additional comments and reflections in the smartphone app at any time. These will be qualitatively analysed to report on the experiences of PwP using the cueing interventions.

**Semi-structured interviews.** We will conduct semi-structured optional interviews with a subsample of participants who consent. The semi-structured interviews will focus on participants experiences of using the cueing methods, experiences of drooling, their preferred cueing method and thoughts on future use of the methods and digital technology. Open ended questions will be utilised for example *"Can you describe your experience of using the CueBand method?"* and *"Can you describe your experience of drooling whilst using this particular method?"*.

### Data analysis

Researchers conducting data analyses will be blinded to the group allocation. Drooling frequency and severity will be analysed before and after the interventions. The primary outcome measure will be changes in the ROMP-S. Secondary outcome measures will include changes in frequency, severity and duration of drooling on the visual analogue scales, the UPDRS 2.2 subset for saliva, NMSQ and PDQ-8. The primary analysis will be a comparison of change from baseline of ROMP-S score for the two intervention groups. Data for the two groups will be compared using standard descriptive statistical analysis (e.g., distribution, central tendency, and dispersion). Mixed effects modelling will be used to compare outcomes with adjustment for crossover and any other relevant covariates where residual differences are not fully accounted for through randomisation. The level of significance will be set at $P<0.05$ and confidence intervals will be reported. The semi-structured interviews will be analysed using Braun and Clark's (2006) reflexive thematic analysis [20].

### Sample size

The sample size has been determined by the difference between the two groups in the change in the ROMP saliva score from baseline to follow up as the primary outcome of interest. However, a minimal clinically important difference (MCID) for this change has not been established. A recent study by Mestre et al., [21] identified an MCID of three (reduction of one point in more than two items) for the ROMP-saliva score based on clinical judgement. In their interventional study the ROMP-saliva scores were; 22.7 (standard deviation (SD) 5.5) at baseline and 16.5 (SD 5.7) at follow-up in the intervention group. The presence of a crossover effect will be investigated and is assumed by a design effect of 1.4, which is likely to be the upper end of any effect. A loss to follow up has been assumed as 30% and all though this seems small, studies involving PwP tend to have relatively low drop-out rates [22]. Setting $\alpha = 0.05$; $\beta = 0.10$ (90% power), MCID at three and assuming the SD for the change in score to be no greater than the SD for the baseline score (5.5), suggests a sample size of 71 per group (142 in total). Allowing for a design effect of 1.4 and loss to follow up of 30%, suggests a minimum sample size of 284.

### Discussion

This study will evaluate a novel device termed CueBand for the management of drooling in PwP. Drooling can lead to adverse negative physical and psychosocial health outcomes and

whilst current treatments decrease the production of saliva, such treatments cause implications for good oral health and digestion. Non-invasive treatment options such as behavioural cueing methods are therefore recommended. In a previous pilot study, we demonstrated that a wearable haptic cueing method was an effective treatment method, and socially acceptable solution for PwP. However, the limitations of the pilot study were the short duration of follow up and small sample size. Longer term follow-up has not yet been detailed and future work is required to test out on larger populations to determine which person, situation, cue type and frequency variables lead to more successful outcomes. While PD-CUE was able to discretely deliver the haptic cueing to the individual, the device itself was cumbersome and was not yet suitable for all day use, with participants wearing it for 1 hour per day only. PD-CUE was rather simplistic in design, with a single toggle switch to start and stop the haptic cueing, and with no ability to modify the intervals or intensity. This required the individual to consciously, and manually, start/stop the cueing as needed and made it a less desirable everyday solution. Moreover, due to the limited functionality, we were unable to capture or understand the real-world behaviours and usage of the intervention to understand the longer-term effects.

This research will address the previous limitations. The new CueBand smartwatch is more discrete and comfortable to wear. It allows individuals to define a 7-day cueing schedule of when the device should start/stop cueing, as well as modify the intervals and intensity of the haptic cues. The inclusion of lifestyle monitoring sensors to CueBand means that we can situate and contextualise the cueing usage alongside the physical activity and sleep behaviours of PwP.

While existing fitness trackers are able to vibrate and provide notifications, they are limited in the control over the vibration motor and those actions are typically driven by the smartphone–i.e. rely on the connection with the smartphone to instruct the fitness tracker to take action. Whereas CueBand has been designed as a standalone solution, meaning that once the device has been programmed with the 7-day cueing schedule, it is possible to then disconnect the smartphone (or leave it at home) and still receive the cueing prompts. All our software is open-source technology and available for anyone to inspect, use or repurpose. Many commercially available wearable devices are closed source or 'black-box' technologies that obfuscate the firmware and algorithms to perform physical activity and sleep monitoring. As such, it is difficult to assess how appropriate those algorithms are for PwP and therefore how accurate the reported data are. The open-source design of CueBand means that any publications or anonymous datasets resulting from the device can be scrutinised and validated against other open-source datasets and algorithms. We will develop a website to publish the anonymous usage data captured from CueBand for future research, alongside our open-source software for the CueBand platform (wearable and mobile apps). By developing this as an open-source platform we allow PwP and researchers to continue to grow and shape CueBand. Being open-source means that anyone can use the tools we've created and develop new functionality or integration with the platform. We will work with Parkinson's UK and Digital Parkinson's (a special interest group for Parkinson's and technology), to establish sustainable processes to enable researchers and PwP alike to access the device and software for personal use and future studies.

Motorically sialorrhea arises from an interaction between oro-facial rigidity, lingual bradykinesia and aspects of oro-pharyngeal dysphagia. Postural, cognitive, attentional and pharmacological factors may also contribute. Objective evaluation of sialorrhea looks at the rate and variability of flow (millilitres or milligrams per unit time; swallow intervals; consistency). Since objective measures seldom reflect patient-reported lived experience, assessment includes rating scales that capture subjective concerns. Evaluation of saliva flow is challenged by a range of issues, including: difficulty obtaining objective measures in naturalistic settings: time and place variability that exists in respect of natural variation in flow rates; fluctuations in motor function experienced by PwP that can impact on swallowing and saliva control; the variety of

situations for PwP concerning where they experience difficulties or not; and the subjective nature of whether an individual perceives there to be a problem present or not (Miller et al. 2019). While objective measures, such as volume of saliva produced, have been used in previous clinical studies they are time consuming and costly and don't lend themselves to research in the community. It is therefore more appropriate to rely on rating scales that this research will utilise. Furthermore, semi-structured interviews are invaluable to explore patients experiences in greater detail and offer insights from the user's perspective.

This research will provide insights into the effectiveness of a novel wrist-worn device for the management of drooling in PwP.

## Supporting information

**S1 Checklist. SPIRIT 2013 checklist: Recommended items to address in a clinical trial protocol and related documents**<sup>∗</sup>.
(DOC)

**S1 File.**
(DOCX)

## Acknowledgments

We would like to acknowledge Parkinson's UK and the participants that will be giving up their time to participate in this study.

## Author Contributions

**Conceptualization:** Kyle Montague, Tiago Guerreiro, Yu Guan, Richard Walker.

**Data curation:** Kyle Montague, Luis Carvalho, Tiago Guerreiro, Dan Jackson, Yu Guan, Richard Walker.

**Formal analysis:** Lorelle Dismore, Kyle Montague, Luis Carvalho, Tiago Guerreiro, Dan Jackson, Yu Guan, Richard Walker.

**Funding acquisition:** Kyle Montague, Tiago Guerreiro, Yu Guan, Richard Walker.

**Investigation:** Lorelle Dismore, Kyle Montague, Luis Carvalho, Tiago Guerreiro, Dan Jackson, Yu Guan, Richard Walker.

**Methodology:** Lorelle Dismore, Kyle Montague, Luis Carvalho, Tiago Guerreiro, Dan Jackson, Yu Guan, Richard Walker.

**Project administration:** Kyle Montague, Luis Carvalho, Tiago Guerreiro, Dan Jackson, Yu Guan, Richard Walker.

**Resources:** Luis Carvalho, Tiago Guerreiro, Dan Jackson, Yu Guan.

**Software:** Kyle Montague, Luis Carvalho, Tiago Guerreiro, Dan Jackson, Yu Guan.

**Supervision:** Richard Walker.

**Writing – original draft:** Lorelle Dismore, Kyle Montague, Tiago Guerreiro, Dan Jackson, Richard Walker.

**Writing – review & editing:** Lorelle Dismore, Kyle Montague, Luis Carvalho, Tiago Guerreiro, Dan Jackson, Yu Guan, Richard Walker.

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
