## [Decision Letter · Decision Letter 0]

31 Oct 2022

PONE-D-22-21557A Protocol for the Evaluation of a Wearable Device for monitoring of symptoms, and cueing for the management of drooling, in people with Parkinson’sPLOS ONE

Thank you for submitting your manuscript to PLOS ONE. After careful consideration, we feel that it has merit but does not fully meet PLOS ONE’s publication criteria as it currently stands. Therefore, we invite you to submit a revised version of the manuscript that addresses the points raised during the review process.

We look forward to receiving your revised manuscript.

Kind regards,

Luigi Lavorgna

Academic Editor

PLOS ONE

Journal Requirements:

Reviewers' comments:

Reviewer's Responses to Questions

**Comments to the Author**

1. Does the manuscript provide a valid rationale for the proposed study, with clearly identified and justified research questions?

Reviewer #1: Yes

Reviewer #2: Yes

2. Is the protocol technically sound and planned in a manner that will lead to a meaningful outcome and allow testing the stated hypotheses?

Reviewer #1: No

Reviewer #2: Yes

3. Is the methodology feasible and described in sufficient detail to allow the work to be replicable?

Reviewer #1: No

Reviewer #2: Yes

4. Have the authors described where all data underlying the findings will be made available when the study is complete?

Reviewer #1: Yes

Reviewer #2: Yes

5. Is the manuscript presented in an intelligible fashion and written in standard English?

Reviewer #1: Yes

Reviewer #2: Yes

6. Review Comments to the Author

You may also provide optional suggestions and comments to authors that they might find helpful in planning their study.

Reviewer #1: Dear Editor,

Thank you for the opportunity to provide a review of Manuscript PONE-D-22-21557 entitled "A Protocol for the Evaluation of a Wearable Device for monitoring of symptoms, and cueing for the management of drooling, in people with Parkinson’s". My comments relate primarily to the adequacy of the implementation and reporting of epidemiologic and statistical procedures.

The quality of the technical English is appropriate, but corrections will be needed. I suggest that the authors undertake a thorough round of copyediting. For example, see Lines 50, 59, 62, and 74.

# Major Issues

## Overall

The authors claim no conflicts of interest, but a device and application are being tested here and the authors describe these using the third person possessive pronoun. It is currently free, but will it always be so? Even if it were free, are there no commercial implications of the resulting data that this trial will produce? What commercialisation options are being explored through this work by the participating universities? The authors need to be more forthcoming about these issues instead of making standard declaration by ticking a box.

Formally, the condition is called "Parkinson's Disease" or "Parkinson Disease". It is most certainly **not** "Parkinson's". This adoption of the colloquial trivialises the condition in my view.

I cannot find a description of the part of the study involving 3,000 participants.

## Participants

I am unclear whether participants will be limited to those in the UK. The website and applications are potentially accessible from around the world. This needs to be clarified.

How will you ensure that participants, especially those with dementia, are mentally competent to provide consent?

Justification for the sample size is missing and must be provided.

## Safety

The authors have not provided information on the exploration of safety or adverse events outcomes. This is an important omission.

## Data analysis

Line 325: What do the authors mean by "crossover"? They must list the covariates that will be used in the adjustments. Alternatively, they must prespecify the definition that they will use to judge whether a covariate is to be used in the adjustment or not.

The authors must specify a level of significance. They must report that they will present confidence intervals.

# Recommendation

I am unable to support the approval of this manuscript for publication in the journal until these issues are considered.

Thank you.

Reviewer #2: Page 4 line 89

Amend: NICE guidelines [11] on the treatment of sleepiness problems recommend that pharmacological or surgical therapy should be preferred over nonpharmacological therapy (such as behavioral stimulus methods).

Page 5 lines 99-113

Redundant concept for an introduction, since then the same thing is explained in the methods. In my opinion, it is better to summarize the basic concepts in a few lines and add these things in the corresponding section.

Also, the introductory section should be shortened.

Further, these reviews could be cited in the introduction as represent extensive summary of digital therapeutics and Parkinson: PMID: 32667839 and PMID: 34018047.

Page 11-12 line 244-265

I would move the "CueBand" section to the beginning, otherwise one reads the entire intervention protocol without knowing the functionality of this tool.

Page 11 line 234

- Are there also exclusion criteria?

- Also, as I understand it, these criteria are only for selecting the 300 out of 3000 to be included in the comparative cueing study. How are the 3000 selected?

Page 11 line 241

For patients recruited through mailing lists, or even those from the clinic, how is it determined that they have "sufficient cognitive function and manual dexterity"?

Page 11 line 244

It does not seem to me to be specified in the text whether the bracelet will also have an auditory signaling method in addition to the vibratory one, which may not be sufficient in some cases.

7. PLOS authors have the option to publish the peer review history of their article (what does this mean?). If published, this will include your full peer review and any attached files.

Reviewer #1: No

Reviewer #2: No

---

## [Author Response · Author response to Decision Letter 0]

15 Dec 2022

We've provided a 'Response to Reviewers' docx file that includes a table with the reviewers comments and our responses.

---

## [Editor Report · Decision Letter 1]

9 Jan 2023

A Protocol for the Evaluation of a Wearable Device for monitoring of symptoms, and cueing for the management of drooling, in people with Parkinson’s Disease

PONE-D-22-21557R1

We’re pleased to inform you that your manuscript has been judged scientifically suitable for publication and will be formally accepted for publication once it meets all outstanding technical requirements.

Kind regards,

Luigi Lavorgna

Academic Editor

PLOS ONE
---

## [Editor Report · Acceptance letter]

13 Feb 2023

PONE-D-22-21557R1 

A Protocol for the Evaluation of a Wearable Device for monitoring of symptoms, and cueing for the management of drooling, in people with Parkinson’s Disease 

Dear Dr. Montague:

I'm pleased to inform you that your manuscript has been deemed suitable for publication in PLOS ONE. Congratulations! Your manuscript is now with our production department. 

Kind regards, 

on behalf of

Dr. Luigi Lavorgna 

Academic Editor

PLOS ONE